# Safety of Four COVID-19 Vaccines across Primary Doses 1, 2, 3 and Booster: A Prospective Cohort Study of Australian Community Pharmacy Vaccinations

**DOI:** 10.3390/vaccines10122017

**Published:** 2022-11-25

**Authors:** Sandra M. Salter, Dani Li, Kevin Trentino, Lisa Nissen, Kenneth Lee, Karin Orlemann, Ian Peters, Kevin Murray, Alan Leeb, Lucy Deng

**Affiliations:** 1School of Allied Health, The University of Western Australia, Perth, WA 6000, Australia; 2MedAdvisor International Pty Ltd., Melbourne, VIC 3000, Australia; 3Medical School, The University of Western Australia, Perth, WA 6000, Australia; 4Centre for Business and Economics of Health, Faculty of Business, The University of Queensland, Brisbane, QLD 4000, Australia; 5SmartVax, Perth, WA 6000, Australia; 6School of Population and Global Health, The University of Western Australia, Perth, WA 6000, Australia; 7Illawarra Medical Centre, Perth, WA 6000, Australia; 8National Centre for Immunisation Research and Surveillance, Westmead, NSW 2145, Australia; 9The University of Sydney Children’s Hospital Westmead Clinical School, Westmead, NSW 2145, Australia

**Keywords:** COVID-19, pharmacy, vaccine, safety, adverse events, surveillance, AstraZeneca, Pfizer, Moderna, Novavax

## Abstract

Four COVID-19 vaccines are approved for use in Australia: Pfizer-BioNTech BNT162b2 (Comirnaty), AstraZeneca ChAdOx1 (Vaxzevria), Moderna mRNA-1273 (Spikevax) and Novavax NVX-CoV2373 (Nuvaxovid). We sought to examine adverse events following immunisation (AEFI) at days 3 and 42 after primary doses 1, 2, 3 and booster. We conducted active vaccine safety surveillance from 130 community pharmacies in Australia integrated with AusVaxSafety, between August 2021–April 2022. Main outcomes: AEFI at 0–3 days post-vaccination; medical review/advice at 3 days and 42 days post-vaccination; SARS-CoV-2 breakthrough infection by day 42. Of 110,024 completed day 3 surveys (43.6% response rate), 50,367 (45.8%) reported any AEFI (highest proportions: Pfizer 42%, primary dose 3; AstraZeneca 58.3%, primary dose 1; Moderna 65.4% and Novavax 58.8%, both primary dose 2). The most common AEFI reported across all doses/vaccines were local reactions, systemic aches and fatigue/tiredness. Overall, 2172/110,024 (2.0%) and 1182/55,329 (2.1%) respondents sought medical review at days 3 and 42, respectively, and 931/42,318 (2.2%) reported breakthrough SARS-CoV-2 infection at day 42. We identified similar AEFI profiles but at lower proportions than previously reported for Pfizer, AstraZeneca, Moderna and Novavax COVID-19 vaccines. Moderna vaccine was the most reactogenic and associated with higher AEFI proportions across primary doses 2, 3, and booster.

## 1. Introduction

The Australian coronavirus disease (COVID-19) vaccination campaign commenced on 22 February 2021 importantly balancing vaccination of priority groups, with vaccine supply constraints, technical capacity (including a skilled workforce), and a system to monitor adverse events following immunisation (AEFI) [1,2]. Two vaccines, Pfizer-BioNTech BNT162b2 (Comirnaty) and AstraZeneca ChAdOx1 (Vaxzevria), were made available through specialised clinics and government vaccination hubs [1,2]. Slow uptake, compounded by vaccine supply issues, safety concerns over the AstraZeneca vaccine, insufficient contracted vaccine administration providers [2], and increasing COVID-19 infection led governments to onboard community pharmacies, as destinations for COVID-19 vaccination from June 2021 [3]. Using an established skilled workforce of pharmacist immunisers (with vaccination rights since 2014), pharmacies could offer AstraZeneca (June 2021), Moderna mRNA-1273 (Spikevax) (September 2021), Pfizer (November 2021), paediatric Pfizer and Moderna (January 2022) and Novavax NVX-CoV2373 (Nuvaxovid) (February 2022) vaccines, with almost 7 million COVID-19 vaccinations administered from Australian pharmacies by 30 April 2022 [4]. Ongoing changes to COVID-19 vaccination schedules and the addition of third primary and booster doses [5] meant pharmacists could administer any of doses 1 through 4, for four different COVID-19 vaccines, in both adults and children.

Clinical trials provide vaccine safety data in limited populations or for limited time periods. Post-marketing surveillance of AEFI is essential to identify reactions arising as whole populations are vaccinated; to maintain public safety and confidence in vaccination; to inform vaccination policy; and to ensure only the safest vaccines remain in use. Spontaneous (passive) surveillance systems may capture severe AEFI but are hindered by under-reporting and can lack timely signal detection. Active systems, such as AusVaxSafety, actively solicit AEFI reports across a defined population via text message or email, and undertake signal detection in near-real time. In the changing COVID-19 vaccination landscape, active surveillance must be linked with vaccine rollout.

Previous research in Australia has examined AEFI reported in the first 3–7 days after one or two primary doses of AstraZeneca and Pfizer vaccines, using data from AusVaxSafety (Australia’s national vaccine safety surveillance system) [6], and from electronic health records [7] in a mass vaccination hub. However, there are no known studies of AEFI reported from pharmacy surveillance; after Moderna and Novavax vaccination (for any dose); or for third primary and booster doses (with any vaccine), in Australia. Uniquely, Australian pharmacy vaccination records distinguish third primary doses from booster doses, offering an opportunity to examine AEFI separately in third primary and booster vaccinations. Pharmacies were initially the only access point for Moderna vaccines in Australia [8]. In order to facilitate safety surveillance of Moderna vaccines, and ensure Australians accessing any COVID-19 vaccination from pharmacy would be included in national vaccine safety surveillance efforts [9], we linked pharmacy vaccinations to AusVaxSafety [10].

Here, we report AEFI recorded via a pharmacy-integrated national vaccine safety surveillance system, over the first nine months of COVID-19 vaccination in Australian pharmacies. Specifically, we sought to examine safety of COVID-19 vaccines at days 3 and 42 after primary doses 1, 2, 3 and booster doses, for AstraZeneca, Moderna, Pfizer and Novavax vaccines.

## 2. Materials and Methods

### 2.1. Study Design, Participants and Setting

We conducted a prospective cohort study of people receiving COVID-19 vaccinations in Australian pharmacies between August 2021 and April 2022. Participants were consecutive individuals who self-selected to receive a COVID-19 vaccination at any of 130 pharmacies participating in active vaccine safety surveillance using a previously described integration [11] linked to AusVaxSafety’s active vaccine surveillance system [9]. Pharmacies were selected across a range of geographic settings, to ensure a representative sample across Australia. Pharmacies were recruited as surveillance sites using an opt-in approach (via direct email and promotion through professional newsletters and Facebook pages) for Western Australia (*n* = 26), and an opt-out approach (each being informed via email that they had been selected for inclusion in active vaccine safety surveillance and this study, with the option to opt out within seven days before first commencing surveillance) for the rest of Australia (*n* = 104). Site and pharmacist immuniser consent were obtained for each pharmacy. An information package was provided to all pharmacies. This included explanatory documents and video for pharmacist immunisers; and posters for display in the vaccination room, to explain the study and AusVaxSafety COVID-19 vaccine safety surveillance to participants. Pharmacies were permitted to leave the study at any time; two pharmacies opted out in September 2021 and a further two opted out in March 2022.

Pharmacists in Australia were variously authorised to administer a range of COVID-19 vaccines to different age groups across different jurisdictions, depending on updates to government policy as Australia’s COVID-19 vaccine rollout progressed [12,13]. COVID-19 vaccines approved in Australia during the study period included: ChAdOx1 nCoV-19 (AZD1222) (Vaxzevria^®^ [AstraZeneca, Cambridge, UK]), BioNTech BNT162b2 (Comirnaty^®^ [Pfizer, and Pfizer Paediatric, New York, NY, USA]), mRNA-1273 (Spikevax^®^ [Moderna and Moderna Paediatric, Cambridge, UK]), and NVX-CoV2373 (Nuvaxovid^®^ [Novavax, Gaithersburg, MD, USA]); hereafter termed by manufacturer name. Participants aged ≥ 5 years were variously eligible over time [12,13,14], for inclusion in this study.

Participant-centred active surveillance was conducted through use of a standardised survey delivered by the SmartVax tool, and developed by AusVaxSafety, Sydney, Australia [6]. All individuals who received a COVID-19 vaccine at a participating pharmacy, and for whom a mobile phone number was recorded, were sent an SMS message by SmartVax after each vaccine dose. The SMS included a direct link to a survey (sent on day 3, day 8 (not included in this analysis), and day 42; only day 3 respondents received the day 42 survey). A reminder SMS was sent within one week if surveys were not completed. Links for the day 3 and 42 surveys expired on days 7 and 49, respectively. Participant consent was included with vaccination consent, with the option to opt-out by not completing the survey when it was sent.

The day 3 survey included questions (with pre-defined response options) about adverse events following immunisation (AEFI), care sought from a healthcare professional for an AEFI (hereafter termed ‘medical review or advice’), actions taken to relieve AEFI (such as fever medication), anaphylaxis history, and chronic medical conditions. The day 42 survey asked if participants had experienced “any illness that needed medical attention”, the type of care sought and diagnosis obtained, and whether they had tested positive for COVID-19. Using survey logic, participants were asked up to 43 questions in the day 3 survey, and up to 5 questions in the day 42 survey, based on responses to each question (Appendix A). Linked participant demographics (age, sex, Australian state or territory where vaccinated) and COVID-19 vaccination details (brand, dose, date of vaccination) were obtained from the pharmacy vaccine encounter record.

### 2.2. Patient and Public Involvement

A community representative was appointed to provide input to protocol development, participant information and consent, and a consumer (adult) perspective to the study overall. In addition, a Community Conversation (CC) [15] on the topic of COVID vaccine safety (for vaccinations administered in pharmacies), was conducted at the commencement of the study. The CC included 5 representatives from the Consumer and Community Involvement Program (CCIP) [15], 13 community members and the research team. Consumers recognised the surveillance survey needed to remain standardised, so no changes to questions were proposed. However, suggestions for improvement included provision of information or resources on AEFI for consumers. An end of survey message with links to information was added mid-way through the study (Appendix A). When pharmacists started vaccinating children, a second community representative was appointed to provide input from the perspective of a parent of young children, noting long-term safety as a priority for this group.

### 2.3. Variables and Data Sources

For day 3 safety, the primary outcome of interest was any AEFI, by COVID-19 vaccine brand and dose, in the first 3 days after vaccination. The secondary outcomes studied were individual adverse event types (local reaction, fever, rash, chills, systemic aches, gastrointestinal symptoms, fatigue or tiredness, fainting/loss of consciousness, seizure/convulsion, other symptoms), and adverse events resulting in medical review or advice. Local reactions were defined as pain, redness, swelling, itching at or near the injection site, while rash referred to a reaction not at injection site. Systemic aches included headache, muscle/body aches, and joint aches/pain. Gastrointestinal symptoms referred to nausea, vomiting, diarrhoea, and/or abdominal pain. In addition, data was available for participant sex, age, COVID-19 vaccine brand, history of chronic medical conditions, anaphylaxis history, and the use of pain or fever medication pre and post vaccination.

For day 42 safety, the primary outcome of interest was any illness needing medical attention in the 42 days after vaccination. Secondary outcomes included type of care sought and self-reported diagnosis.

Data for the analysis were obtained from the SmartVax system [16]. Participant demographic and vaccination data received from participating pharmacies, via an integration with MedAdvisor as previously described [11], were combined with survey responses collected by the SmartVax tool.

### 2.4. Statistical Methods

All statistical analyses were conducted using R (version 4.2.1). Numbers and percentages of vaccinations with at least one AEFI were reported. Separate multivariable logistic regression models for each COVID-19 vaccination dose were fitted to analyse characteristics associated with the response: reporting of any AEFI at day 3. These models included sex, age, history of chronic disease, history of anaphylaxis, the use of fever or pain medication before vaccination, and the vaccine brand as covariates.

In the day 42 survey participants who answered “yes” to the question “Have you had any illness that needed medical attention?” were prompted to enter their diagnosis to a free text field. Diagnoses were coded in accordance with MedDRA^®^, the Medical Dictionary for Regulatory Activities terminology, which is the international medical terminology developed under the auspice of the International Council for Harmonisation of Technical Requirements for Pharmaceuticals for Human Use [17]. Diagnoses were coded in duplicate, based on MedDRA’s lowest level terms (LLT) or preferred terms (PT) and are reported descriptively as preferred terms.

### 2.5. Ethics Approval

This study was approved by The University of Western Australia Human Research Ethics Committee (2019/RA/4/20/5907). Results are reported according to STROBE checklist for cohort studies [18].

## 3. Results

The characteristics of participants receiving a COVID-19 vaccination are presented in Table 1, and Appendix A. Between August 2021 and April 2022, 256,733 vaccinations were administered in participating pharmacies. Of these, 47,870 (18.6%) were first dose, 75,914 (29.6%) were second dose, 5277 (2.1%) were third primary dose, and 127,672 (49.7%) were a booster dose. The most common COVID-19 vaccination administered was Moderna 137,233 (53.5%), followed by Pfizer 85,691 (33.4%), AstraZeneca 30,480 (11.9%) and Novavax 3329 (1.3%). Of vaccinations administered, 34.3% were to females and 32.0% to males; 33.7% had no sex recorded. The median age of participants receiving Pfizer vaccine (33 years) was lower than those receiving Moderna (39 years), Novavax (43 years) and AstraZeneca (46 years). Specifically, 45.5% of Pfizer primary doses 1–3 were in participants aged under 12; 99.5% of Pfizer booster doses were in participants aged over 16; and 99.9% of all Moderna vaccinations were in participants aged over 16 years. Over 80% of participating pharmacy vaccinations were administered in the Australian States of Victoria (100,352), New South Wales (59,037) and Western Australia (52,794). Of survey responses, 2224 (2.0%) reported a history of anaphylaxis and 12,241 (11.1%) reported at least one chronic medical condition. Of these, the most common chronic medical conditions were diabetes (3084/12,241; 25.1%), chronic inflammatory conditions (2088/12,241; 17.1%), heart disease (1612/12,241; 13.2%), and obesity (1203/12,241; 9.8%) (Appendix A). ‘Other’ chronic medical conditions were reported by 4558/12,241 (37.2%) participants

### 3.1. Adverse Events Day 3 after Vaccination

A total 110,024 (43.6%) responses to the day 3 surveys were received. Of these, 50,367 (45.8%) reported any AEFI, with the highest proportion (65.4%) reported after the second dose Moderna vaccination. Overall, 32,124 (29.2%) of participants took pain or fever medication (as pre-medication) at the time of vaccination, although this varied by vaccine and dose. Proportions of AEFI reported after third primary and booster doses were similar for both AstraZeneca (22.6% dose 3; 23.4% booster) and Pfizer vaccines (42% dose 3; 41.6% booster). A higher proportion of participants receiving Moderna third primary dose reported AEFI, than those receiving a booster dose (61.3% vs. 54.7%, respectively). Of those reporting an AEFI, 29,478 (58.5%) used a treatment (such as pain/fever medication, icepack/cream, or antihistamine medication) after vaccination, to relieve symptoms (Table 1). Across all vaccination brands and doses the three most common symptoms reported were local reactions (pain, redness, swelling, itching at or near the injection site); systemic aches (headache, muscle/body aches, or joint aches/pain); and fatigue or tiredness (Figure 1).

Of the 110,024 respondents to the day 3 survey, 2172 (2.0%) reported seeking medical review or advice for an AEFI. Participants aged 30–39 represented the highest proportion of those seeking medical review or advice across all vaccines and doses, except for AstraZeneca booster dose and Pfizer third primary dose (in which the highest proportion of medical review or advice was reported in those aged 40–49). Multiple care types were reported: phone advice from a health service (*n* = 763), care from a general practitioner or Aboriginal healthcare worker (*n* = 937), and visiting a hospital emergency department (*n* = 340) (Table 1; Appendix A).

Respondents receiving Moderna or Pfizer vaccines were more likely to report any adverse event following their second (Moderna: 65.4%, Pfizer: 31.6%), third (Moderna: 61.3%, Pfizer: 42.0%), and booster vaccinations (Moderna: 54.7%, Pfizer: 41.6%) when compared to their first vaccination (Moderna: 39.7%, Pfizer: 26.1%). In contrast, respondents receiving the AstraZeneca vaccine were several times more likely to report any adverse event following their first vaccination when compared to their second, third, and booster vaccinations (first dose: 58.3%, second dose: 23.3%, third dose: 22.6%, booster: 23.4%). This pattern was consistent across the individual AEFI presented in Figure 1.

#### Characteristics Associated with Reporting an Adverse Event—Day 3

The vaccination brand, respondent age, sex, underlying chronic condition, and anaphylaxis history were associated with the reporting of any AEFI on day three across all COVID-19 vaccine doses (Table 2). Respondents with underlying chronic conditions, those with a history of anaphylaxis, females, and those in the 30–39 and 40–49 age brackets were consistently more likely to report adverse events following all doses of COVID-19 vaccinations received. Respondents over 80 years of age were consistently less likely to report AEFI.

### 3.2. Adverse Events Day 42 after Vaccination

A total of 98,760 day 42 surveys were sent, of which 55,329 (56.0%) were completed. Overall, 1182 (2.1%) of respondents reported having any illness that needed medical attention by day 42 following vaccination (Table 1). Of those reporting any illness by day 42, 1050 (88.8%) responded to the question regarding the type of medical care sought. Multiple care types were reported (participants could select multiple care types): general practitioner, Aboriginal healthcare worker or specialist (*n* = 747); visiting a hospital emergency department (*n* = 289); and admission to hospital (*n* = 147). Overall, 42,318 responses to the day 42 question ‘Have you tested positive for COVID-19 since you received your COVID-19 vaccine?’ were received. Of these, 931 (2.2%) responded ‘yes’. The breakdown by vaccination brand and dose is provided in Table 1.

Information on the diagnosis of the new illness was provided in 1109/1182 (94.1%) of surveys. These free-text diagnoses mapped to 243 MedDRA terms, and three non-MedDRA terms. The top 22 terms (comprising 19 MedDRA and three non-MedDRA terms) accounted for 50% of all reported diagnoses. Of the MedDRA terms, COVID-19 infection was the most reported diagnosis, in 66 (6.0%) respondents. A total 89 (8.0%) respondents reported either lower respiratory chest infection (*n* = 26), viral infection (*n* = 23), urinary tract infection (*n* = 21), or pneumonia (*n* = 19); 65 (5.9%) reported either chest pain (*n* = 36; 26 following Moderna), pericarditis (*n* = 16; 15 following Moderna) or myocarditis (*n* = 13; 10 following Moderna); 36 (3.2%) reported musculoskeletal pain; 21 (1.9%) reported herpes zoster and 21 (1.9%) reported headache. Of diagnoses that did not map to MedDRA terms, 36 (3.2%) were ‘awaiting diagnosis’; and 32 (2.9%) had a ‘self-declared vaccine reaction’. See Appendix A for all reported diagnoses

## 4. Discussion

In this analysis we examine the safety of four COVID-19 vaccines (AstraZeneca, Moderna, Novavax and Pfizer), across primary doses 1, 2, 3 and booster, using data collected via active surveillance of more than a quarter-million pharmacy vaccinations in Australia. Community pharmacies progressively offered all four vaccines (including paediatric formulations), for people aged 5 years and over, in accordance with changing legislation and government approvals for the COVID-19 vaccine rollout in Australia [1,2].

We extend the evidence for safety at day 3 for Pfizer and AstraZeneca vaccines, and provide initial evidence for Moderna and Novavax vaccines, particularly for doses beyond the primary 2-dose schedule.

We found similar proportions of AEFI reported after AstraZeneca doses 1 (58%) and 2 (23%) to clinical trial (43% and 25%) [19] and other real-world evidence (44 and 27%) [20]. We found similar proportions of AEFI after Pfizer dose 1 (26%) and 2 (31%), compared to a large study from Jordan (32% and 32%) [20], but substantially fewer AEFI compared to other post-marketing research (between 45–92%) [21,22,23], and phase 2/3 trials (between 66–83%) [24]. Notably, AusVaxSafety surveillance of the first six months of the COVID-19 vaccine rollout in Australia showed similar proportions of AEFI reported after AstraZeneca doses 1 and 2 (52% and 22%), but substantially higher AEFI reported after Pfizer doses 1 and 2 (35% and 54%) [6], however these results are for adults, whereas 58% of the participants receiving Pfizer dose 1, and 42% receiving dose 2 in our study were aged under 12 years. By comparison, children aged 5–11 years participating in active surveillance in the United States, reported more AEFI than our overall proportions for both Pfizer dose 1 (35–55%) and dose 2 (41–58%) [25]. Reassuringly at every dose point, children < 12 years and adults > 80 years were the least likely to report AEFI in our study.

A higher proportion in our study reported AEFI after Moderna dose 1 (39%) and dose 2 (65%) compared to the other vaccines, however this was still lower than in previous postmarketing and clinical trial research (80–89%) [21,22,26,27]. Novavax was provisionally approved for use in Australia in January 2022 [28] (and recommended as a booster for certain groups in March 2022) [29]; from this study we report Novavax data for January-April 2022. Given the short timeframe under which it has been used in Australia and only early use of Novavax globally [30] there is scant evidence of postmarketing surveillance. However, our study showed a slightly lower proportion of AEFI after Novavax dose 1 (35%) and dose 2 (58%) vaccination compared to clinical trial data (45% and 64%) [31]. Overall, Moderna vaccinations showed the highest reactogenicity: people in our research were more likely to report AEFI after Moderna vaccination (for doses 2, 3 and booster), compared to AstraZeneca, Pfizer and Novavax vaccines, consistent with past research [21,32,33]. Whether this is a feature of Moderna vaccine itself, or a response to recency of vaccine use in the broader population remains to be seen. Concerns of myocarditis and pericarditis with mRNA vaccines may have driven reporting of any AEFI to Pfizer and Moderna vaccines: for Pfizer this early window of reporting was before our study, whereas for Moderna, it was at the start of our study.

Uniquely, our pharmacy data differentiates third primary doses (in Australia, for people with specific immunocompromising conditions and therapies) [34], from third doses given as a booster. In our study, Moderna vaccines showed higher reactogenicity in the third primary than the booster dose, whereas for Pfizer vaccines reactogenicity increased with each progressive dose to the third dose, regardless of whether it was a third primary or booster. This may be due to dose amount: for Moderna, third primary doses are a full dose whereas booster doses are a half dose, whereas all other vaccines are given as full doses across the vaccination schedule [5].

Sustained lower AEFI across all vaccines and doses observed in our study compared to Australian and international research may be due to several factors. Our response rate to the day 3 survey (43%) is lower than response rates of 56–79% reported in other AusVaxSafety surveillance [6], and people with AEFIs may have chosen not to report. However, participants in our study represented the general population receiving vaccinations later in the rollout, as opposed to the priority groups and healthcare workers who received their vaccinations at the start of Australia’s vaccination program [6], when sentiment to respond to surveillance surveys was high. Our study population was exposed to repeated government messaging and global reports of vaccination campaigns to normalise and promote COVID-19 vaccination, and may have been more comfortable accepting minor reactions as normal. Other reasons for our lower AEFI include removal of vaccine mandates (and potentially less vaccination of people fearful of COVID-19 vaccines); differences in patient engagement (our study was opt-out, whereas, other active (V-Safe [35] and ZOE-COVID [36]) and passive (VAERS [37]) surveillance programs reporting high rates of AEFI are all opt-in); or a true effect—that over time the population perceive or experience less AEFI worthy of reporting. Of interest, pharmacists vaccinated children aged under 12 years from the commencement of their access to COVID-19 vaccines in Australia. Despite the vaccine being new in this group, our proportions overall were still lower than previous estimates. A recent systematic review of placebo control groups in COVID-19 vaccine clinical trials found the profile of solicited AEFI (such as those in our survey) was the same as those reported in treatment groups (albeit lower), suggesting a substantial proportion of AEFI are due to nocebo effects [38]. This may explain why the proportions of AEFI reported in our study, from August 2021 to April 2022, were lower (although similar in type), to research conducted earlier in the pandemic.

Interestingly in previous research, we found people reported significantly fewer AEFI after influenza vaccination in Australian pharmacies compared to vaccination by other providers [11]. Seemingly being vaccinated in a pharmacy reduces medicalisation of the process of vaccination. ‘Walk-ins’—people obtaining vaccination without a prior appointment—remain popular in community pharmacies, as they provide timely and convenient access to vaccination for local communities [39]. It may be that people being vaccinated in pharmacies are less inclined to focus on the procedure or its after-effects, and instead view their vaccination as something routine. Regardless, our results highlight the importance of undertaking vaccine safety surveillance from a range of destinations in order to reflect the breadth of vaccination experiences, and better estimate true AEFI.

Overall, 2.0% of respondents reported seeking medical review or advice in the first 3 days after vaccination (more than in previous research [6]), with a higher proportion seeking review or advice after AstraZeneca dose 1 (4.7%) and Moderna dose 2 (4.6%). This may be consistent with reactogenicity profiles: reports of medical advice or review were higher across all vaccines and doses, when proportions for reporting any AEFI were higher. Consistent with previous AusVaxSafety data, medical review or advice was reported more by people aged 30–39 (across almost all vaccines and doses) than any other age group [6]. Beyond medical review or advice, this age group were also consistently more likely to report any AEFI.

Similarly, 2.1% of respondents reported having a new illness that needed medical care, in the first 42 days after vaccination. As with reported medical review or advice in the first three days, care at day 42 was most frequently sought from a general practitioner. Self-reported diagnoses mapped to 243 MedDRA terms with COVID-19 diagnosis being the most reported new illness, however diagnoses were not verified, thus may not be causally linked to the vaccination, and all mapped terms should be interpreted with caution. Indeed, a large proportion of those who reported seeking medical care for a new illness were unable to provide a diagnosis, indicating instead ‘unknown or unidentified’ illness. These may represent nocebo responses, where individuals experience adverse symptoms largely driven by the expectation that a reaction will or has occurred [40]. It should be noted that where dose schedules were less than 42 days (e.g., for m-RNA vaccines), participants did not receive a day 42 survey for that particular dose. Furthermore, the booster dose day 42 surveys could represent cumulative or long-term impacts of repeated dosing. For the m-RNA vaccines, in which we had a high number of booster doses administered, it is reassuring that less than 2% of respondents reported any new illness needing medical care at day 42, after booster dosing.

Finally, we collected self-reported SARS-CoV-2 breakthrough infection data at day 42 after vaccination. The percentage of reported breakthrough infection (after booster dosing) was highest after AstraZeneca (4.3%), compared to Moderna (2.8%) and Pfizer vaccination (3.4%). For immunocompromised people receiving third primary doses the reported breakthrough percentages were higher for all vaccines (6.8%, 4.4% and 4.5%, respectively). These figures offer insight and sit within those reported elsewhere [41], but are limited by lack of diagnostic verification. Furthermore, these percentages may have been different for different parts of Australia. At the time of this study, different jurisdictions had border closures both within Australia and to the rest of the world, which would have impacted COVID-19 infection rates. In Western Australia particularly, background COVID-19 infection rates were extremely low due to hard borders which isolated the state from April 2020 to March 2022 [42]. Further, emerging new variants, including the shift from Delta to Omicron as the dominant SARS-CoV-2 strain during our study, may have led to reduced vaccine effectiveness. Moderna and Pfizer bivalent vaccines were provisionally approved in Australia for booster dosing (including via pharmacies) in October 2022 [43,44], and COVID-19 vaccines are recommended for co-administration with other vaccines (such as influenza [45]). As our COVID-19 vaccination programs transition from pandemic response to ‘business as usual’, ongoing/seasonal active vaccine safety surveillance of new and updated vaccines remains essential to ensure safety and public confidence.

### Limitations

Active vaccine safety surveillance requires people vaccinated to respond to post-vaccination surveys regardless of whether they experience an AEFI or not. Respondent bias is a feature of such surveillance: certain people may be more motivated to report AEFI, while those without AEFI may not understand the importance of providing a ‘no’ response. Higher proportions of AEFI were reported after dose 2 for Pfizer, Moderna and Novavax vaccines, yet these doses had the lowest response rates of all day 3 surveys, suggesting respondent bias existed at least at this dose, and that true AEFI proportions and patterns may be lower than observed.

The survey was only available to those sent a link via SMS. A rise in SMS scams, especially those with links to malware, prompted government warnings to consumers to avoid opening links sent from unknown numbers [46]. This may have increased suspicion as to the veracity of our link sent via SMS, and reduced survey response rates, including in people who experienced AEFI. Beyond this, people without access to a smart phone or the Internet (for example many homeless or elderly people) could not participate. We did not examine cultural or language features of respondents, and recognise different levels of health literacy may impact interpretation of solicited AEFI listed in the survey. Moreover, the survey was only presented in English, which could have impacted engagement. Similarly, survey access for children under 16 years was via the mobile phone number recorded at the pharmacy, and it was not possible to determine who completed the survey (parent, other carer or child). AEFIs reported in this instance may have been missed or subjectively biased.

Participants were sent up to six SMS text messages (including day 3, day 8 (not reported here), day 42 initial and reminder messages) per vaccine dose. SMS fatigue may have resulted in lower engagement, or may have motivated people to report an AEFI that they may not have otherwise reported, had they not received so many messages. Pandemic fatigue, frustration with governments and vaccine mandates may also have impacted survey responses. Similarly, the SMS schedule—in which day 42 surveys were only sent to day 3 respondents may have both biased AEFI reporting at day 42 and distorted survey response rates.

We did not verify AEFIs reported by participants, and recognise this limitation on certainty of AEFI diagnosis, including for solicited reactions. Furthermore, it is difficult to review and interpret free text (‘other’) fields from large-scale data (as in active surveillance, where millions of records are collected [8]). In our study this included both AEFI and chronic medical conditions, and although we mapped free text AEFI terms to MedDRA, we did not use clinical coders, nor undertake verification of free text AEFI reports. Future and ongoing analysis could consider use of machine learning and artificial intelligence (as being developed in the UK [47]) to interrogate free text data, and add insights beyond static surveys. Further, although the proportion of participants reporting they sought medical review or advice was similar at day 3 and day 42 we did not verify such attendance, and these reports may also have been subject to respondent bias.

Policy changes meant COVID-19 vaccine dose schedules, eligibility and access to vaccines changed during the rollout, meaning any of the vaccines and doses 1, 2, 3, or booster may have been given under different conditions for different people. Further (particularly as our study progressed), people could choose which brand of vaccine they wished to receive, and to attend any site for vaccination (for example, mass vaccination hub, GP, pharmacy or others). Not every vaccination site is included in active surveillance, and doses therefore could not be linked for every individual. As a result, each person’s vaccination sequence is unknown, and the extent to which this may have impacted AEFI remains to be examined. This is important at a vaccine level (given different platforms) and also for individual age groups (given different AEFI reporting rates): elderly people received mostly AstraZeneca followed by m-RNA vaccines, whereas younger people mostly received a mix of m-RNA vaccines. We analysed each dose separately and accordingly present likelihood of reporting AEFI using odds ratios, however had we been able to analyse multiple doses within each individual, we would have required a different statistical method. While we could not analyse heterologous vs. homologous dose schedules, or impact of time across the vaccination sequence or between doses, our results are nonetheless indicative of real-world experience, at a time that included mandatory vaccination with available vaccines, and when Australia was opening borders and moving to a new-normal in the pandemic.

## 5. Conclusions

The use of active vaccine safety surveillance from community pharmacies enabled timely monitoring of existing (AstraZeneca and Pfizer) and additional (Moderna and Novavax) COVID-19 vaccines, across a four-dose schedule during the 2021–2022 Australian vaccination campaign. This real-world study identified lower proportions of solicited AEFI than previously reported, for all four vaccines, particularly for primary first and second doses. We add new evidence for AEFI following primary third primary and booster doses, with Moderna vaccinations associated with higher proportions of AEFI across primary doses 2 and 3, and booster doses, compared to Pfizer, AstraZeneca and Novavax vaccines. In early results for Novavax, the proportion reporting AEFI is similar to that observed after Pfizer vaccination, however more people reported seeking medical review or advice after Novavax than any other vaccine. Our results suggest a nocebo effect may exist in surveillance of newly introduced COVID-19 vaccines. Continuing to examine AEFI over time, and allowing for vaccine acceptance with maturation in reporting, may provide more realistic AEFI estimates.

## Figures and Tables

**Figure 1 vaccines-10-02017-f001:**
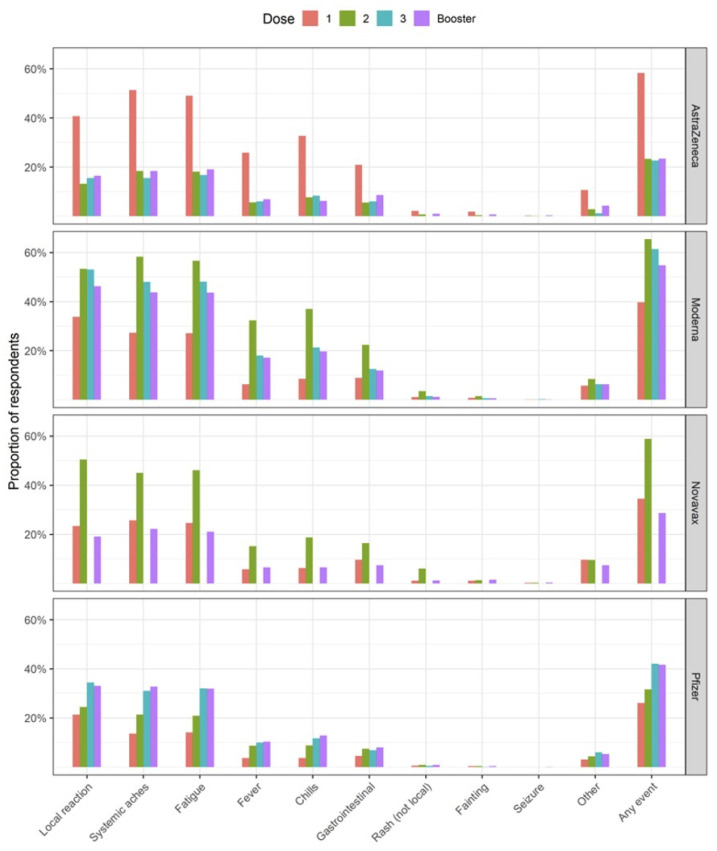
Proportion of individual adverse events (%) reported on day 3 following COVID-19 vaccination, stratified by vaccination brand and dose. Local reaction was defined as pain, redness, swelling, itching at or near the injection site. Systemic aches included headache, muscle/body aches, and joint aches/pain. Gastrointestinal included nausea, vomiting, diarrhoea, and/or abdominal pain. Six participants receiving Novavax as dose 3 responded to the day 3 survey, of which 0 events were reported.

**Table 1 vaccines-10-02017-t001:** Characteristics of COVID-19 vaccination encounters by vaccine brand and dose number.

	AstraZeneca				Moderna				Novavax				Pfizer			
	Dose 1	Dose 2	Dose 3	Booster	Dose 1	Dose 2	Dose 3	Booster	Dose 1	Dose 2	Dose 3	Booster	Dose 1	Dose 2	Dose 3	Booster
n	4076	25,587	156	661	29,870	37,508	2218	67,637	1386	1248	14	681	12,538	11,571	2889	58,693
Vaccinated age (years), median (Q1, Q3)	47 (32, 63)	45 (29, 63)	59 (40, 68)	58 (42, 67)	32 (20, 48)	32 (19, 47)	53 (36, 65)	47 (32, 62)	41 (31, 55)	43 (32, 58)	61 (51.5, 74)	48 (34, 62)	11 (8, 23)	13 (9, 31)	42 (30, 58)	39 (27, 57)
Sex																
Female	1293 (31.7)	7725 (30.2)	58 (37.2)	218 (33.0)	10,850 (36.3)	12,939 (34.5)	812 (36.6)	24,328 (36.0)	439 (31.7)	430 (34.5)	4 (28.6)	265 (38.9)	3745 (29.9)	3609 (31.2)	940 (32.5)	20,323 (34.6)
Male	1661 (40.8)	9628 (37.6)	49 (31.4)	205 (31.0)	11,609 (38.9)	13,822 (36.9)	659 (29.7)	19,635 (29.0)	353 (25.5)	327 (26.2)	3 (21.4)	211 (31.0)	3643 (29.1)	3582 (31.0)	783 (27.1)	16,014 (27.3)
Not recorded	1122 (27.5)	8234 (32.2)	49 (31.4)	238 (36.0)	7411 (24.8)	10,747 (28.7)	747 (33.7)	23,674 (35.0)	594 (42.9)	491 (39.3)	7 (50.0)	205 (30.1)	5150 (41.1)	4380 (37.9)	1166 (40.4)	22,356 (38.1)
Chronic medical condition/s	271 (15.6)	1163 (12.4)	19 (22.6)	62 (20.4)	1328 (9.7)	1243 (8.9)	357 (31.1)	4051 (12.8)	100 (15.6)	84 (15.7)	3 (50.0)	63 (19.4)	260 (4.5)	221 (5.0)	300 (21.4)	2716 (10.9)
History of anaphylaxis	51 (2.9)	140 (1.5)	4 (4.8)	12 (3.9)	331 (2.4)	321 (2.3)	27 (2.3)	624 (2.0)	20 (3.1)	15 (2.8)	0 (0.0)	13 (4.0)	105 (1.8)	99 (2.2)	18 (1.3)	444 (1.8)
Pain/fever medicine pre vaccination	597 (34.5)	1793 (19.1)	13 (15.5)	48 (15.8)	3082 (22.6)	5750 (41.2)	332 (28.9)	10,235 (32.4)	111 (17.3)	120 (22.4)	3 (50.0)	61 (18.8)	1322 (22.7)	1087 (24.4)	381 (27.2)	7189 (28.8)
Day 3 Survey																
Sent	3901 (95.7)	25,407 (99.3)	154 (98.7)	650 (98.3)	29,489 (98.7)	37,354 (99.6)	2133 (96.2)	66,326 (98.1)	1338 (96.5)	1214 (97.3)	11 (78.6)	614 (90.2)	12,398 (98.9)	11,345 (98.0)	2776 (96.1)	57,362 (97.7)
Responded	1732 (44.4)	9410 (37.0)	84 (54.5)	304 (46.8)	13,647 (46.3)	13,944 (37.3)	1149 (53.9)	31,592 (47.6)	643 (48.1)	536 (44.2)	6 (54.5)	324 (52.8)	5818 (46.9)	4460 (39.3)	1399 (50.4)	24,976 (43.5)
Responded age (years), median (Q1, Q3)	51 (35, 64)	53 (33, 64)	59.5 (50.25, 66.25)	60 (48, 68)	34 (19, 50)	35 (18, 51)	56 (42, 67)	52 (35, 64)	43 (32, 57)	46 (35, 59)	68.5 (57.25, 76.75)	52.5 (40, 67)	10 (8, 17)	11 (8, 27)	48 (34, 61)	45 (29, 60)
Reported adverse event	1010 (58.3)	2196 (23.3)	19 (22.6)	71 (23.4)	5423 (39.7)	9120 (65.4)	704 (61.3)	17,292 (54.7)	222 (34.5)	315 (58.8)	0 (0.0)	93 (28.7)	1516 (26.1)	1408 (31.6)	587 (42.0)	10,391 (41.6)
Medication to relieve symptoms	634 (63.0)	1183 (54.2)	13 (68.4)	35 (49.3)	2555 (47.4)	6089 (67.1)	446 (63.6)	10,497 (61.0)	110 (50.0)	185 (59.3)	0 (0.0)	46 (49.5)	724 (48.0)	774 (55.3)	313 (53.5)	5874 (56.9)
Reported medical review or advice ^a^	81 (4.7)	107 (1.1)	0 (0.0)	6 (2.0)	225 (1.6)	648 (4.6)	13 (1.1)	554 (1.8)	21 (3.3)	16 (3.0)	0 (0.0)	12 (3.7)	57 (1.0)	73 (1.6)	15 (1.1)	344 (1.4)
Phone advice	36 (46.2)	30 (34.5)	0 (0.0)	2 (50.0)	86 (42.6)	238 (40.4)	2 (28.6)	184 (41.3)	8 (44.4)	8 (50.0)	0 (0.0)	5 (45.5)	21 (42.9)	29 (44.6)	3 (27.3)	111 (39.6)
Care from a GP	38 (48.7)	52 (59.8)	0 (0.0)	1 (25.0)	97 (48.0)	294 (49.9)	5 (71.4)	221 (49.7)	10 (55.6)	9 (56.2)	0 (0.0)	4 (36.4)	26 (53.1)	28 (43.1)	8 (72.7)	144 (51.4)
Emergency department visit	15 (19.2)	16 (18.4)	0 (0.0)	1 (25.0)	45 (22.3)	119 (20.2)	1 (14.3)	71 (16.0)	3 (16.7)	2 (12.5)	0 (0.0)	2 (18.2)	12 (24.5)	12 (18.5)	0 (0.0)	41 (14.6)
Day 42 Survey																
Sent	1731 (42.5)	9399 (36.7)	82 (52.6)	236 (35.7)	13,600 (45.5)	13,875 (37.0)	1052 (47.4)	27,656 (40.9)	469 (33.8)	184 (14.7)	0 (0.0)	45 (6.6)	5654 (45.1)	2856 (24.7)	1269 (43.9)	20,652 (35.2)
Responded	999 (57.7)	5487 (58.4)	47 (57.3)	151 (64.0)	7482 (55.0)	7574 (54.6)	687 (65.3)	15,976 (57.8)	300 (64.0)	150 (81.5)	0 (0.0)	35 (77.8)	2911 (51.5)	1616 (56.6)	779 (61.4)	11,135 (53.9)
Responded age (years), median (Q1, Q3)	58 (41, 65)	60 (39, 66)	63 (57, 69.5)	62 (56, 69)	37 (20, 53)	39 (19, 55)	59 (47.5, 68)	56 (40, 66)	47.5 (34, 60)	48 (37.25, 57.75)	-	58 (42, 68)	10 (8, 13)	12 (9, 35)	52 (40, 64)	51 (35, 63)
Reported new illness	40 (4.0)	113 (2.1)	2 (4.3)	5 (3.3)	192 (2.6)	213 (2.8)	20 (2.9)	300 (1.9)	19 (6.3)	3 (2.0)	0 (0.0)	3 (8.6)	62 (2.1)	33 (2.0)	13 (1.7)	164 (1.5)
Tested COVID-19 positive	9 (0.9)	36 (0.7)	3 (6.8)	3 (4.3)	24 (0.3)	72 (1.0)	27 (4.4)	273 (2.8)	0 (0.0)	0 (0.0)	0 (0.0)	0 (0.0)	199 (7.6)	37 (5.1)	29 (4.5)	219 (3.4)

All reported as n (%). ^a^ Respondents could select multiple levels of medical review or advice. This shows engagement with the health system, not the highest level of care sought. Proportions add to more than 100%. Age is shown overall, at day 3 and at day 42 for comparison of those vaccinated with those who responded.

**Table 2 vaccines-10-02017-t002:** Characteristics associated with the reporting of any event on day 3 following COVID-19 vaccinations at pharmacies. Values presented are adjusted odds ratios (95% CI).

	Primary Dose 1	Primary Dose 2	Primary Dose 3	Booster
Vaccine brand				
AstraZeneca	Reference	Reference	Reference	Reference
Moderna	0.49 (0.44, 0.55)	5.28 (4.94, 5.64)	5.18 (3.00, 8.95)	3.46 (2.61, 4.57)
Novavax	0.37 (0.30, 0.45)	4.56 (3.79, 5.50)	*	1.17 (0.80, 1.70)
Pfizer	0.36 (0.31, 0.41)	2.04 (1.83, 2.27)	2.15 (1.25, 3.70)	1.93 (1.46, 2.55)
Age group (years)				
Under 12	0.56 (0.49, 0.65)	0.47 (0.41, 0.55)	0.61 (0.05, 6.90)	0.29 (0.16, 0.54)
12–15	0.79 (0.71, 0.88)	0.96 (0.87, 1.07)	1.14 (0.25, 5.28)	0.67 (0.45, 1.01)
16–19	0.64 (0.56, 0.74)	0.69 (0.61, 0.78)	0.72 (0.40, 1.29)	0.85 (0.78, 0.93)
20–29	Reference	Reference	Reference	Reference
30–39	1.26 (1.13, 1.39)	1.25 (1.13, 1.38)	1.27 (0.92, 1.77)	1.09 (1.02, 1.16)
40–49	1.21 (1.09, 1.35)	1.23 (1.11, 1.36)	1.40 (1.02, 1.92)	1.08 (1.01, 1.15)
50–59	0.92 (0.82, 1.04)	0.95 (0.85, 1.05)	1.07 (0.78, 1.46)	0.90 (0.84, 0.95)
60–69	0.76 (0.67, 0.86)	0.84 (0.75, 0.92)	0.86 (0.63, 1.17)	0.67 (0.63, 0.71)
70–79	0.57 (0.47, 0.68)	0.65 (0.56, 0.75)	0.73 (0.51, 1.05)	0.48 (0.45, 0.52)
80+	0.28 (0.20, 0.39)	0.35 (0.27, 0.45)	0.46 (0.26, 0.79)	0.29 (0.26, 0.33)
Sex				
Male	Reference	Reference	Reference	Reference
Female	1.43 (1.33, 1.53)	1.46 (1.37, 1.56)	2.03 (1.65, 2.51)	1.72 (1.65, 1.80)
Not recorded	1.23 (1.14, 1.33)	1.22 (1.14, 1.31)	1.52 (1.23, 1.89)	1.36 (1.30, 1.43)
Chronic medical condition/s	1.70 (1.53, 1.88)	1.48 (1.34, 1.62)	1.27 (1.04, 1.55)	1.53 (1.45, 1.62)
History of anaphylaxis	1.35 (1.12, 1.63)	1.44 (1.18, 1.74)	1.74 (0.91, 3.33)	1.30 (1.14, 1.48)
Pain/fever medicine pre vaccination	2.45 (2.29, 2.62)	3.26 (3.08, 3.46)	2.76 (2.29, 3.34)	3.05 (2.93, 3.17)

* Responses following dose 3 of the Novavax vaccine were excluded due to low numbers (*n* = 14).

## Data Availability

Not available as we did not seek ethical approval for the release/sharing of data.

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
