# Peer review of "Safety of Four COVID-19 Vaccines across Primary Doses 1, 2, 3 and Booster: A Prospective Cohort Study of Australian Community Pharmacy Vaccinations"

_vaccines, 2022, doi:10.3390/vaccines10122017_

Round 1
Reviewer 1 Report
Dear authors,
I read with interest your work
The topic is important and the manuscript is well organisated.
But before considering for publisher it, I have some minor comments
1. I suggest adding a flow-chart of your study
2. In your study, 256,733 vaccinations were administered. But how many participants were included? Were any patients followed up after multiple injections in your study? If yes, how many? and if so, I recommend using the generalized linear latent and mixed model for statistical analysis. Because of logistic regression is not appropriate in this condition
Author Response
"Please see the attachment."

Reviewer 2 Report
Overview and general recommendation:
In the manuscript, the authors test the adverse events following immunisation (AEFI) at day4 and day32 after dose1,2,3 and booster of four COVID-19 vaccines which are in use in Australia, namely Pfizer-BioNTech BNT162b2 (Comirnaty), AstraZeneca ChAdOx1 (Vaxzevria), Moderna mRNA-1273 (Spikevax) and Novavax NVX-CoV2373 (Nuvaxovid). They generate AEFI reports for different doses of different vaccines at different time points. In this study, all of the four vaccines show lower AEFI compared with previous postmarketing and clinical research. And Moderna vaccine results in the highest reactogenicity of all four vaccines.
I find the paper is organized in a proper way and most of the results are well described. The authors should add the importance of AEFI data in background. And major methods are well described in the manuscript and properly used in the research. And the conclusion is fully supported by the data.
Major comments:
1. The authors mainly analysis the AEFI of adults. I think the authors can also analysis the data of children in the discussion and conclusion part.
2. In introduction part, I suggest the authors explain why AEFI data is important for a vaccine. They can introduce how AEFI report is generated and how can we evaluate a vaccine through an AEFI report.
Author Response
"Please see the attachment."

Round 2
Reviewer 1 Report
Dear Authors,
Thank you for your responses
About my suggestion that you didn't solve, please include it in the Limitation section
Author Response
"Please see the attachment."
